# The Silent Thought: Modeling Internal Cognition in Full-Duplex Spoken Dialogue Models via Latent Reasoning

**Donghang Wu** [* 1]  **Tianyu Zhang** [* 2]  **Yuxin Li** [1]  **Hexin Liu** [1]  **Chen Chen** [1]  **Eng Siong Chng** [1]  **Yoshua Bengio** [2]

## Abstract

During conversational interactions, humans subconsciously engage in concurrent thinking while listening to a speaker. Although this internal cognitive processing may not always manifest as explicit linguistic structures, it is instrumental in formulating high-quality responses. Inspired by this cognitive phenomenon, we propose a novel **F**ull-duplex **LA**tent and **I**nternal **R**easoning method named FLAIR that conducts *latent* thinking simultaneously with speech perception. Unlike conventional "thinking" mechanisms in NLP, which require post-hoc generation, our approach aligns seamlessly with spoken dialogue systems: during the user's speaking phase, it recursively feeds the latent embedding output from the previous step into the next step, enabling continuous reasoning that strictly adheres to causality without introducing additional latency. To enable this latent reasoning, we design an Evidence Lower Bound-based objective that supports efficient supervised finetuning via teacher forcing, circumventing the need for explicit reasoning annotations. Experiments demonstrate the effectiveness of this think-while-listening design, which achieves competitive results on a range of speech benchmarks. Furthermore, FLAIR robustly handles conversational dynamics and attains competitive performance on full-duplex interaction metrics.

## 1. Introduction

Speech serves as the most natural and intuitive interface for human communication, offering a bandwidth-efficient and emotionally resonant medium for interaction. Con-

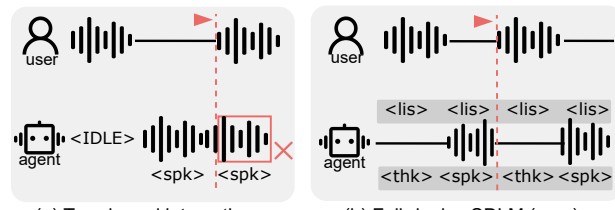

*Figure 1.* (a) Turn-based interaction. The agent remains idle and starts to respond after the end of the user turn, but cannot be interrupted by user, as shown in the red box. (b) Full-duplex SDLM continuously listens to *streaming* speech input, supports user barge-in, and automatically switches between thinking and speaking like a human speaker.

sequently, enabling machines to understand and generate speech with human-like proficiency has long been a central goal in Human-Computer Interaction. In recent years, driven by large-scale data (Cui et al., 2024) and deep learning techniques, end-to-end Spoken Dialogue Language Models (SDLMs) have emerged as a focal point in both academia and industry (Nguyen et al., 2023), demonstrating significant improvements in general intelligence and the naturalness of conversational dynamics.

Despite these advancements, a fundamental discrepancy remains between current model architectures and the cognitive reality of human communication. Most existing SDLMs inherit the sequential execution patterns of text-based Large Language Models (LLMs) (Chu et al., 2024), where the acoustic user query and agent response are flattened into a single sequence and processed via autoregressive next-token prediction (Brown et al., 2020). This serialized approach starkly contrasts with human physiology, where auditory perception and vocal expression operate as independent, concurrent systems—enabling us to listen and speak simultaneously. This capability, known as full-duplex interaction in dialogue models, has recently emerged as a key research frontier in speech and NLP (Défossez et al., 2024; Yu et al., 2025). By continually ingesting streaming user speech during response prediction, a speaking-while-listening agent delivers more natural, fluid, and human-like conversations, with the ability to take turns proactively, offer backchannel responses, and gracefully yield to user barge-ins (Wu et al., 2025).

---
[*]Equal contribution [1]College of Computing and Data Science, Nanyang Technological University [2]Mila, Quebec Artificial Intelligence Institute, Université de Montréal. Correspondence to: Chen Chen <chen1436@e.ntu.edu.sg>.

*Proceedings of the 43rd International Conference on Machine Learning*, Seoul, South Korea. PMLR 306, 2026. Copyright 2026 by the author(s).

This paradigm shift raises a fundamental open research question: how should an "always-on" model optimally utilize its computational capacity when the user is talking? Early full-duplex implementations typically defaulted to predicting non-informative padding tokens while listening (Défossez et al., 2024; Hu et al., 2025). While functional, these repeated tokens contribute nothing to the generation of the subsequent response. A seemingly more plausible alternative is to adapt the Chain-of-Thought (CoT) mechanism (Wei et al., 2022) from NLP by generating explicit textual reasoning sequences in parallel with the input audio. However, this approach is problematic for streaming speech interaction. Since reasoning cannot causally precede the unfolding user speech, generating discrete text tokens creates a synchronization mismatch. Furthermore, the unpredictable nature of spoken turn-taking implies that a user may finish speaking at any moment; if the model is locked into generating a rigid textual thought chain, terminating this process to switch to response generation introduces latency and state-management complexities. Consequently, we propose a more radical trajectory: can we abandon explicit, token-based thinking entirely in favor of implicit reasoning? We hypothesize that maintaining a dynamic, information-rich latent state that evolves continuously with the input can effectively accumulate reasoning cues to enhance the quality of the next response.

To realize this concept, we formulate the concurrent reasoning process as a latent-variable modeling problem, addressing the challenge that "internal thoughts" lack explicit supervision. We propose a novel **F**ull-duplex **LA**tent and **I**nternal **R**easoning (FLAIR) method with a training framework grounded in variational inference, specifically optimizing the Evidence Lower Bound (ELBO) (Zhang et al., 2018). Our method introduces a non-causal Global-aware Expert during the training phase, which leverages the full dialogue context to derive an approximate posterior distribution of optimal latent embeddings. The causal SDLM is then trained to minimize the Kullback–Leibler (KL) divergence (Hershey & Olsen, 2007) between its generated latent prior and the expert's posterior. This alignment effectively transfers global reasoning capability to the causal model, enabling it to perform continuous, implicit thinking via dense embeddings based on streaming user input. Extensive empirical evaluations demonstrate that our proposed paradigm significantly enhances response quality across a diverse spectrum of tasks, ranging from factual knowledge and multi-turn Question Answering (QA) to open-ended generation and multiple-choice benchmarks. Crucially, our latent reasoning mechanism achieves these gains without requiring additional supervisory data or incurring any inference latency. Consequently, the model preserves the high efficiency and robustness essential for real-time conversational dynamics, maintaining both low response latency and high interaction success rates.

The contributions of this paper are summarized as follows:

- We propose a fully causal latent reasoning method that empowers full-duplex SDLMs with "think-while-listening" capabilities. This approach eliminates the need to construct explicit reasoning datasets, strictly adheres to causality constraints during the listening phase, and introduces no additional computational overhead during inference.

- We introduce an efficient supervised fine-tuning (SFT) strategy for latent reasoning. This method circumvents the limitations of existing approaches by enabling effective optimization via teacher forcing. Furthermore, our approach is orthogonal to existing latent-reasoning optimization techniques. Reinforcement learning can be seamlessly applied as a post-training step to further improve model performance.

- This work is the first to introduce latent reasoning into the domain of speech LLMs. While we implement our method within a full-duplex SDLM framework due to its critical demand for "think-while-listening" capabilities, the proposed approach can be readily adapted to traditional half-duplex speech LLMs, offering valuable insights for the future design of speech LLMs.

## 2. Related Work

**Latent Reasoning** Recent work on the latent chain of thought challenges the assumption that complex reasoning must be externalized as discrete, language-based tokens (Shalev et al., 2024; Wang et al., 2023a; Turpin et al., 2023). Instead, it treats reasoning as iterative computation in continuous hidden states, reducing reliance on long textual traces and potentially improving efficiency (Zhu et al., 2025b;a). Coconut and CODI (Hao et al., 2024; Shen et al., 2025) feeds the model's last layer hidden state back as the next input embedding so the model can explore alternatives in latent space rather than committing early to a single textual trajectory. Ouro and looped transformer (Zhu et al., 2025c; Nikunj Saunshi and Nishanth Dikkala and Zhiyuan Li and Sanjiv Kumar and Sashank J. Reddi, 2025) propose looped language models that internalize multi-step reasoning during pretraining through repeated latent updates and learned depth allocation, and report strong gains that appear to reflect improved knowledge manipulation. Soft Thinking, SoftCoT++ (Zhang et al., 2025b; Xu et al., 2025b;a) complements these training-based approaches with a training-free mechanism that generates soft concept states as continuous mixtures of token embeddings, enabling implicit exploration over multiple candidate next steps while keeping outputs interpretable.

**Full-Duplex Spoken Dialogue Systems** Spoken interaction is transitioning from conventional turn-based paradigms (Sarikaya et al., 2002) toward advanced full-duplex architectures (Veluri et al., 2024; Wang et al., 2024a) that support concurrent listening and speaking. Current research in this domain prioritizes engineering components such as streaming ASR (Moritz et al., 2020), incremental text-to-speech (TTS) (Chiba & Higashinaka, 2025), robust barge-in management (Schlangen & Skantze, 2011), and incremental dialogue strategies (Zhang et al., 2025a). Beyond cascaded pipelines, recent work has also explored end-to-end training of full-duplex SDLMs (Défossez et al., 2024; Yu et al., 2025). For example, Yu et al. propose SALMONN-omni (Yu et al., 2025), the first codec-free full-duplex speech LLM enabling the model to autonomously decide when to listen, speak, and yield, while preserving text-centric reasoning and achieving strong spoken QA and conversational performance. However, a significant limitation of existing frameworks is that during the user's turn, they typically enforce silence by repeatedly outputting pause tokens without performing any reasoning. Several approaches implement a streaming "think-while-listening" process by constructing explicit streaming CoTs (Wu et al., 2025; Chiang et al., 2025a; Arora et al., 2025). These approaches either require complex procedures to build specialized CoT datasets (Wu et al., 2025; Chiang et al., 2025a) or fail to guarantee the causality of the CoT process (Arora et al., 2025). Our proposed FLAIR framework addresses this gap by enabling continuous cognitive activity during the listening phase.

**Reasoning in Language Models** Large language models achieve complex reasoning through intermediate trajectories, such as chain-of-thought (CoT) (Wei et al., 2022), self-consistency (Wang et al., 2023b), and tree-of-thoughts (Yao et al., 2023). Specialized paradigms such as program-aided language models (PAL) (Gao et al., 2023) and Toolformer (Schick et al., 2023) further incorporate structured or externalized reasoning steps. Nevertheless, these strategies are primarily designed for static text settings and require the full problem context before reasoning can begin. Because they rely on non-causal computation and hypothesis revision, such "stop-and-think" mechanisms are fundamentally incompatible with the strict causality and low-latency requirements of real-time conversational streaming.

## 3. Preliminary

### 3.1. Full-duplex SDLMs

Full-duplex SDLMs represent a paradigm shift from conventional turn-based interactions to systems that continuously perceive user speech streams while generating responses in real time, enabling agents to handle dynamic behaviors such as backchanneling and user barge-in.

A typical full-duplex SDLM architecture consists of a streaming speech encoder, an LLM backbone, and a speech generation module. The streaming encoder produces continuous embeddings $X \in \mathbb{R}^{T \times D}$, where $T$ denotes the number of frames and $D$ denotes the feature dimension. The LLM backbone (parameterized by $\theta$) autoregressively predicts the agent's text tokens $Y^{\text{txt}}$, which can be written as:

$$P_\theta(Y^{txt}|X) = \prod_{t=1}^{T} P_\theta(Y_t^{txt}|Y_{<t}^{txt}, X_{\leq t}). \quad (1)$$

Finally, an autoregressive decoder (parameterized by $\psi$) predicts the agent's speech tokens $Y^{\text{spc}}$ based on the text tokens and the hidden states $h^l$ from the LLM:

$$P_\psi(Y^{spc}|X, Y^{txt}, h^l) =$$
$$\prod_{t=1}^{T} P_\psi(Y_t^{spc}|Y_{<t}^{spc}, Y_{\leq t}^{txt}, h_{\leq t}^l). \quad (2)$$

In conventional full-duplex systems, the LLM is required to predict silence tokens `<SIL>` to fill the agent's output stream during the listening phase. The expected agent text token sequence in the $i$-th turn is structured as:

$$\begin{aligned} Y^{txt} = [&< \texttt{SIL} >, \ldots, < \texttt{SIL} >, \\ &< \texttt{BOS} >, R_{i,1}, R_{i_2}, \ldots, R_{i,T}, \quad (3) \\ &< \texttt{PAD} >, \ldots, < \texttt{PAD} >, < \texttt{EOS} >], \end{aligned}$$

where $R_{i,t}$ is the response token at time step $t$, and `<BOS>` and `<EOS>` denote the beginning and the end of the agent's turn, respectively. The `<PAD>` token is used to pad the gap between the text and speech channels to ensure that they reach the same length. While this mechanism facilitates full-duplex interaction, it leaves the listening window unexploited and may degrade performance due to repetitive generation of a single token. The objective of this paper is to optimize the assistant's output while the user is speaking, thereby achieving higher-quality response content, i.e., higher-quality $Y^{\text{txt}}$.

### 3.2. Evidence Lower Bound

In Latent Variable Models, maximizing the marginal likelihood $\log P(x)$ is intractable due to the high-dimensional integration required over the latent space. To address this, Variational Inference introduces a parameterized distribution $q_\phi(z|x)$ for latent variable $z$ to approximate the true posterior (Zhang et al., 2018). The ELBO serves as a surrogate optimization objective, derived via Jensen's Inequality to provide a functional lower bound on the log-likelihood:

$$\log P(x) \geq \mathbb{E}_{q_\phi(z|x)}[\log P(x|z)] - \text{KL}[q_\phi(z|x)||P(z)]. \quad (4)$$

This objective operates through two competing mechanisms: the first term maximizes the expected reconstruction likelihood to ensure that the latent variables capture essential

data features, while the second term minimizes the KL divergence to regularize the approximate posterior toward a prior distribution $P(z)$ (Kingma & Welling, 2013; Higgins et al., 2017). This dual optimization ensures both generative fidelity and a well-structured latent space. A comprehensive mathematical derivation of this bound is provided in Appendix A.

# 4. Methodology

Unlike conventional full-duplex methods mentioned in Section 3.1, which predict silence tokens while the user is speaking, this paper employs latent reasoning to realize a "think-while-listening" mechanism for the full-duplex model, as depicted in Figure 2. During the user's speaking phase, the input to the LLM at the current step is no longer the token output from the previous step but a continuous embedding. Inspired by (Zeng et al., 2025; Zhang et al., 2025b), we do not directly adopt the previous hidden state $h_{t-1}^l \in \mathbb{R}^d$ as the input for the current step $t$. Instead, the hidden state $h_{t-1}^l$ is processed through the LLM head to generate text logits $y_{t-1}^{\text{txt}} \in \mathbb{R}^{|V|}$, where $V$ is the vocabulary with size $|V|$. We then use $y_{t-1}^{\text{txt}}$ to compute a weighted average over the LLM vocabulary to obtain the input embedding $Z_{t-1}$ for step $t$. This process can be formulated as: $Z_{t-1} = \text{Softmax}(y_{t-1}^{txt})E$, where $E \in \mathbb{R}^{|V| \times d}$ denotes the vocabulary embedding matrix of the LLM.

This latent reasoning approach precludes effective teacher-forcing supervision during the training phase, as latent representations cannot be obtained to supervise the LLM's output. This paper proposes an ELBO-based method that employs a non-causal "Global-aware Expert" model, $Q_\phi$, to derive the approximate posterior of $Z$ by extracting latent features from the entire dialogue session. This section first introduces the variants of the ELBO in Full-duplex SDLMs, and subsequently describes the implementation of training Full-duplex SDLMs for latent reasoning grounded in ELBO principles.

## 4.1. ELBO in Full-duplex SDLMs

Unlike standard latent variable models that focus on the reconstruction of input data in Section 3.2, our objective is to maximize the conditional likelihood $P_\theta(Y^{txt}|X)$. To optimize the conditional log-likelihood, which is generally intractable due to the integration over the latent space $Z$, we employ Variational Inference by introducing an approximate posterior $q_\phi(Z|X, Y^{txt})$. Following the principles of the ELBO, the objective can be derived as follows:

$$
\begin{aligned}
\log P_\theta(Y^{txt}|X) \geq \mathbb{E}_{q_\phi(Z|X,Y^{txt})}[\log P_\theta(Y^{txt}|Z,X)]- \\
\text{KL}[q_\phi(Z|X,Y^{txt})||P_\theta(Z|X)].
\end{aligned}
\tag{5}
$$

The derivation is provided in Appendix B. This formulation effectively shifts the model's focus from data reconstruction to intent-based response generation.

The resulting conditional ELBO consists of two components that drive the latent reasoning capability of the SDLMs: (1) Conditional Reconstruction, which encourages the model to generate a high-quality response $Y^{txt}$ given the observed user input $X$ and the latent reasoning $Z$, following a posterior distribution $q_\phi(Z|X, Y^{txt})$ offered by a "Global-aware Expert" model; (2) Variational Regularization, which acts as a constraint that forces the causal model's latent prior $P_\theta(Z|X)$ to align with the Global-aware Expert model's posterior distribution $q_\phi(Z|X, Y^{txt})$, guiding the SDLM to internalize reasoning capabilities within its latent space. A more detailed explanation is provided in Appendix B.

## 4.2. SFT for Latent Reasoning

Based on the aforementioned ELBO principle, we construct an SFT method employing teacher forcing. We first define a latent reasoning timing label $G \in \mathbb{R}^T$. Specifically, $G_t = 1$ when the time step $t$ corresponds to the assistant's turn (i.e., $t$ falls between the indices of <BOS> and <EOS>), and $G_t = 0$ otherwise. The complete user speech stream and assistant text token stream $Y^{txt}$, which is defined in (3), are converted into speech embeddings $X$ and text embeddings $H^{txt}$, respectively. These are summed and fed into a non-causal encoder, termed the Global-aware Expert model $Q_\phi$. This Expert model leverages global conversational information to derive the ideal latent reasoning embedding labels. To address the distributional discrepancy between the LLM's input and output spaces, we project the Expert's output hidden representation $h_t^e$ via a linear layer to the vocabulary dimension. Subsequently, a Softmax function is applied to obtain the vocabulary weighting distribution $W_t^e \in \mathbb{R}^{|V|}$. The ideal latent reasoning embedding label $Z_t$ is then obtained by weighting the vocabulary embedding matrix $E$ with $W_t^e$. This process can be formulated as:

$$
h^e = Q_\phi(X + H^{txt}),
\tag{6}
$$

$$
W_t^e = \text{Softmax}(\text{Linear}(h_t^e)),
\tag{7}
$$

$$
Z_t = W_t^e E.
\tag{8}
$$

We expect $Q_\phi$ to learn the approximate posterior distribution $q_\phi(Z|X, Y^{txt})$ when it has access to the entire dialogue. After deriving the latent reasoning embedding label, the LLM's input embeddings are defined as follows:

$$
H_{in} = X + (1 - G) \odot Z + G \odot H^{txt},
\tag{9}
$$

This means that we use the latent reasoning embedding labels to replace the embeddings of <SIL> tokens used as padding during the user's speech.

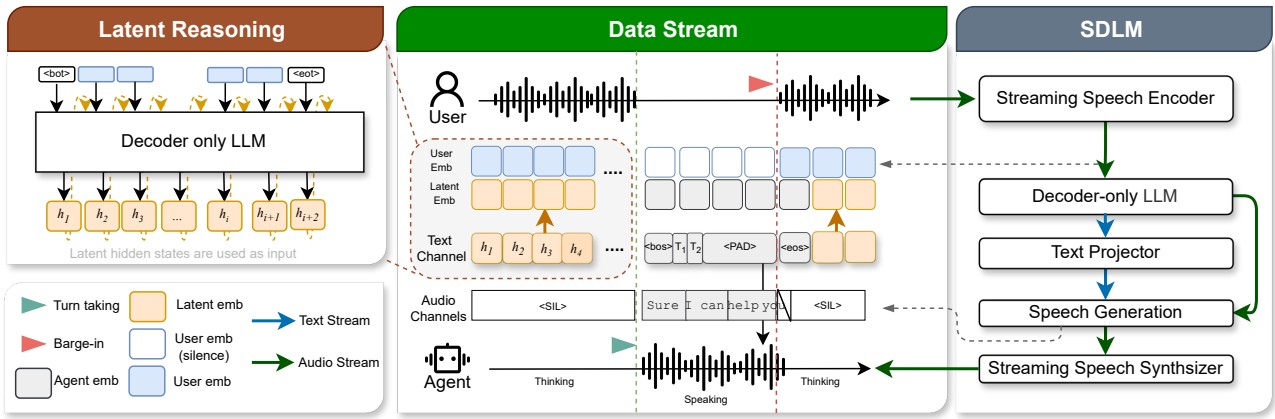

*Figure 2.* The overview of proposed FLAIR. During the user's speech phase, the LLM performs latent reasoning, using the LLM's output latent embeddings as the input for the next step. Once the user finishes speaking, the assistant autonomously decides when to respond; the LLM then executes an explicit forward pass, using text tokens as the input for the next step. When the user barges in, the LLM autonomously decides when to stop speaking and reverts to a state of latent reasoning.

Consequently, we can implement efficient latent reasoning SFT with teacher forcing by utilizing both the latent reasoning embedding labels and the text token labels. Upon feeding $H_{in}$ into the LLM backbone, we obtain the last hidden state $h^l$. This state is then passed through the LLM head to predict the text logits $\hat{y}^{txt}$. By applying the argmax operation over these logits, we obtain the predicted text tokens $\hat{Y}^{txt}$. Inspired by (5), the loss function for ELBO-based latent reasoning training comprises two components. The first component is the conditional reconstruction loss, specifically the next-token prediction loss for the response segment, defined as:

$$
\begin{aligned}
\mathcal{L}_{reco} &= -\log P_\theta(Y^{txt} \mid X, Z) \\
&= -\sum_{t=1}^{T} G_t \log P_\theta\left(y_t^{txt} \mid X_{\leq t}, Z_{\leq t}, y_{<t}^{txt}\right).
\end{aligned} \tag{10}
$$

We use the latent reasoning timing label $G$ as a mask so that the loss is computed only when the assistant is speaking. The second component is the variational regularization loss, defined as:

$$
\mathcal{L}_{regu} = \sum_{t=1}^{T} (1 - G_t) \operatorname{KL}\left(\operatorname{sg}[W_t^e] \parallel \operatorname{Softmax}\left(\hat{y}_t^{txt}\right)\right). \tag{11}
$$

Here, sg[·] denotes the stop-gradient operator. We use the vocabulary-weighting distribution produced by the Global-aware Expert model, namely, the weights used to compute the latent-reasoning embedding labels, to supervise the vocabulary-weighting distribution predicted by the LLM during the user's speaking phase (i.e., the listening phase). The latent reasoning timing label $G$ is similarly applied to ensure that this loss is computed exclusively during the user's speaking phase.

Through the conditional reconstruction loss, we guide the Expert model to infer an informative posterior distribution that supports accurate response generation. Meanwhile, via the variational regularization loss, we encourage the LLM to learn a prior that aligns with this posterior during the listening phase. Finally, the last hidden state $h^l$ is passed through an additional multi-layer perceptron (MLP) to obtain $\tilde{G}$, which predicts the latent reasoning timing indicator. This prediction governs the control logic at inference time, determining whether to perform latent reasoning or to generate explicit response tokens. The corresponding loss is defined as:

$$
\mathcal{L}_{time} = -\sum_{t=1}^{T} \left[G_t \log \hat{G}_t + (1 - G_t) \log(1 - \hat{G}_t)\right]. \tag{12}
$$

In summary, the total loss for ELBO-based SFT for latent reasoning is:

$$
\mathcal{L}_{elbo} = \mathcal{L}_{reco} + \alpha \cdot \mathcal{L}_{regu} + \beta \cdot \mathcal{L}_{time}, \tag{13}
$$

where $\alpha$ and $\beta$ are the weights of the loss terms, chosen based on empirical experiments.

During inference, we discard the Global-aware Expert model and perform step-by-step decoding, as illustrated in Figure 2. When $\hat{G}_t$ is predicted to be 1, we output the text token and feed its embedding as the input at the next step. When $\hat{G}_t$ is predicted to be 0, we use the text logits to compute a weighted sum over the vocabulary embedding matrix, yielding a latent-reasoning embedding that serves as the next-step input; meanwhile, a <SIL> token is passed to the speech generation module. The speech generation module continuously consumes text-token inputs and generates speech tokens in a streaming manner. A streaming flow-matching model then converts these speech tokens into speech (Du et al., 2024).

# 5. Data Synthesis and Model Training

## 5.1. Dialogue Data Generation

Existing spoken dialogue datasets suffer from inherent limitations in both scale and domain suitability. They predominantly consist of casual, peer-to-peer interactions (e.g., Fisher (Cieri et al., 2004)), which are insufficient for training an agent to function as a helpful, instruction-following assistant. To overcome this data scarcity, we construct a large-scale synthetic dataset with user and agent streams by integrating LLMs with TTS systems equipped with voice cloning capabilities. We categorize training data into three subsets: 530K hours of **speech continuation** data, 70k hours of **instruction-following QA** data, and 20k hours of **ASR-QA** data (from a real ASR corpus with background noise). Data statistics, the generation toolkit, and configurations are provided in the Appendix C.

To mitigate the risk of the model overfitting to synthetic acoustic artifacts, we implement a robust diversification strategy that includes various data augmentation techniques, an extensive speaker pool, and different TTS models, as illustrated in Appendix C.

## 5.2. Training pipeline

We categorize the training process into three distinct stages: Pre-training, Latent Reasoning SFT, and Speech Synthesizing SFT. During the Pre-training phase, we employ the conventional full-duplex approach without latent reasoning, training the LLM with speech continuation data to understand speech inputs and establish real-time conversational capabilities. The Latent Reasoning SFT phase is conducted on all data and further divided into two sub-stages: In the first sub-stage, we utilize $\mathcal{L}_{reco}$ exclusively as the loss function to enable the Global-aware Expert model to learn to generate appropriate latent reasoning labels. In the second sub-stage, we employ the complete $\mathcal{L}_{elbo}$ as the objective, training the LLM and the Expert model jointly, aligning the LLM's output with the latent reasoning embeddings generated by the Expert model. Consequently, we obtain an LLM equipped with latent reasoning capabilities, capable of delivering higher-quality text responses. Finally, in the Speech Synthesizing SFT phase, we freeze all parameters excluding the speech generation module and utilize the same speech dataset to train the streaming TTS model (Du et al., 2024).

We employ the same speech dataset utilized in the Latent Reasoning SFT stage, training the speech generation module to accurately generate speech tokens conditioned on the text token labels and LLM hidden states. Specifically, we extract speech tokens using the audio codec from (Du et al., 2024) to serve as training labels and optimize the model via the next-token prediction objective (Brown et al., 2020). Finally, the predicted speech tokens are converted into speech using the corresponding streaming flow matching model in (Du et al., 2024). More training details are in Appendix D.

# 6. Evaluation and Benchmark

The evaluation of SDLM are in three aspects: response accuracy and quality, conversational behavior, and synthesized speech quality. (1) We utilize four factual knowledge test sets in quick QA format, including Llama Questions (Nachmani et al., 2023), WebQuestions (Berant et al., 2013), TriviaQA (Joshi et al., 2017), and SDQA (Faisal et al., 2021). Furthermore, we use two open-ended QA test sets AlpacaEval and CommonEval (real human speech) from VoiceBench (Chen et al., 2024) to verify the instruction following capabilities. We then select two multiple-choice question test sets to assess the speech understanding and reasoning capabilities, which are OpenbookQA and MMSU (Wang et al., 2025). Open-ended QA is evaluated by GPT score on a scale of 1 to 5 (where 1 represents the worst and 5 represents the best), and others are reported by accuracy, where the prompt and criteria follow the official VoiceBench Github. We use Whisper-large-v3 to transcribe the generated speech into text for calculating evaluation metrics (Radford et al., 2022) (2) To assess the conversational dynamics of our full-duplex SDLM, we adopt the evaluation framework proposed by (Chen et al., 2025). We report three key metrics: Turn-taking latency, Barge-in latency, and Barge-in success rate. For these evaluations, we utilize the *impatient* dataset from (Chen et al., 2025), characterized by frequent interruptions occurring at approximately 2-second intervals. We also evaluate the performance on Full-Duplex-Bench (Lin et al., 2025), covering the model's turn-taking capability on the real-world noisy CANDOR dataset (Reece et al., 2023), as well as its barge-in performance on synthetic dialogue data. For the CANDOR dataset, the metrics include the Takeover Rate (TOR) and the turn-taking latency. For the barge-in scenarios, we evaluate the assistant response's TOR, GPT-4o score (0-5 scale), and latency, upon the conclusion of the user's speech after an interruption (3) We report the MOS score calculated by UTMOS (Saeki et al., 2022) on the synthesized speech.

# 7. Result and Analysis

## 7.1. Reasoning Abilities

We first evaluate the performance of our approach on several commonly used QA benchmarks to verify the improvements in response quality brought by latent reasoning. We compare our method with existing classical full-duplex models as well as various half-duplex models. Additionally, we include a comparison with STITCH (Chiang et al., 2025b), which achieves low-latency real-time responses through a "think-while-speaking" mechanism. Since existing "think-

*Table 1.* Accuracy (%) or GPT-Score of different methods on QA benchmarks. Results of baseline systems are taken from (Yu et al., 2025; Chen et al., 2024). We use "*thk*" to denote the proposed latent reasoning. "FD", "LlamaQ", "WebQ", "TriQA", "OBQA", "AlpacaE" and "ComE" stand for "if Full-duplex", Llama Questions, WebQuestions, TriviaQA, OpenbookQA and AlpacaEval, CommonEval respectively.

| Method | FD | LlamaQ | WebQ | TriQA | SDQA | AlpacaE | ComE | OBQA | MMSU |
|---|---|---|---|---|---|---|---|---|---|
| Moshi (Défossez et al., 2024) | ✓ | 54.5 | 22.1 | 16.7 | 15.6 | 2.01 | 1.60 | 25.9 | 24.0 |
| Freeze-Omni (Wang et al., 2024b) | ✓ | 56.2 | 27.9 | 28.5 | 53.5 | 4.03 | 3.46 | 31.0 | 28.1 |
| SALMONN-omni (Yu et al., 2025) | ✓ | 73.6 | 43.7 | 56.0 | - | 3.22 | - | - | 30.0 |
| SALM-Duplex (Hu et al., 2025) | ✓ | 51.3 | 25.0 | 16.9 | 26.0 | 2.99 | 2.50 | 39.6 | 26.3 |
| GLM-4-Voice (Zeng et al., 2024) | ✗ | 65.7 | 37.0 | 47.5 | 37.0 | 3.97 | 3.42 | 53.4 | 39.8 |
| Qwen2-Audio (Chu et al., 2024) | ✗ | 69.7 | 45.2 | 40.3 | 35.7 | 3.74 | 3.43 | 49.5 | 35.7 |
| Kimi-Audio (Ding et al., 2025) | ✗ | 68.3 | 37.3 | 51.2 | 63.1 | 4.46 | 3.97 | 83.5 | 62.2 |
| Baichuan-Audio (Li et al., 2025) | ✗ | 74.0 | 40.7 | 53.0 | 45.8 | 4.41 | 4.08 | 71.7 | 53.2 |
| STITCH-R (Chiang et al., 2025b) | ✗ | 70.0 | 50.3 | 49.6 | - | 2.70 | - | - | - |
| FLAIR w/o *thk* | ✓ | 73.0 | 41.7 | **53.8** | 54.4 | 3.80 | 3.54 | 72.9 | 50.2 |
| FLAIR w/ *thk* | ✓ | **78.0** | **43.0** | 51.2 | **56.2** | **3.85** | **3.65** | **74.2** | **56.2** |

*Table 2.* Evaluation of conversational behaviors of different methods on the *Impatient* dataset proposed in (Chen et al., 2025), in terms of turn-taking latency (s), barge-in latency (s), and barge-in success rate (%). We use "*thk*" to denote the proposed latent reasoning.

| Method | E2E | Turn Taking | Barge-in | | MOS |
|---|---|---|---|---|---|
| | | Latency (↓) | Latency (↓) | Success rate (↑) | |
| Freeze-Omni (Wang et al., 2024b) | ✗ | 1.17 | 1.20 | 79.5 | 4.3 |
| dGSLM (Nguyen et al., 2023) | ✓ | 0.57 | 0.86 | 85.0 | 2.2 |
| Moshi (Défossez et al., 2024) | ✓ | - | 0.81 | 55.1 | 3.9 |
| ORISE (Chen et al., 2025) | ✓ | 0.43 | 0.61 | 96.8 | 4.2 |
| FLAIR w/o *thk* | ✓ | **0.33** | 0.49 | **100** | **4.3** |
| FLAIR w/ *thk* | ✓ | 0.39 | **0.46** | **100** | **4.3** |

while-listening" methods either lack open-source code, CoT datasets, and commonly used benchmark results, or are restricted to specific application scenarios (Wu et al., 2025; Chiang et al., 2025a; Shih et al., 2025; Arora et al., 2025), they are excluded from this comparison.

Table 1 presents the performance across different tasks. It is evident that enabling the SDLM to perform latent reasoning during the user's speaking phase yields performance gains across nearly all tasks compared to the baseline without latent reasoning. The improvements are particularly significant in tasks requiring understanding and reasoning, such as MMSU. The gains are relatively smaller on tasks like Web Questions, as factual knowledge extraction requires less reasoning. These results demonstrate the efficacy of latent reasoning in enhancing the reasoning capabilities of full-duplex systems.

Furthermore, our approach outperforms existing full-duplex SDLMs in most of the tasks, especially on MMSU and OpenbookQA. Although certain half-duplex models outperform our method on specific benchmarks, we attribute this to two primary factors: (1) There is a low similarity between our training data and the benchmark data, as our data construction process does not explicitly target the specific scenarios represented in these benchmarks. (2) Our model is trained on conversational data, where assistant re-

sponses tend to be relatively concise for daily interactions. In contrast, half-duplex models typically generate longer responses. For open-ended QA tasks such as AlpacaEval and CommonEval, longer responses tend to yield a higher GPT score. As shown in Table 5, our method generates shorter responses. However, compared to the text LLM's response, our method provides responses that are more concise yet more effective and suitable for spoken agents. Despite these factors, our method still achieves superior results compared to half-duplex models across numerous benchmarks, which proves the effectiveness of our architecture and training pipeline. Finally, metrics on CommonEval, which is a real-human speech dataset, confirm that our method remains effective on real human speech, demonstrating its robustness.

### 7.2. Conversational Behavior

Table 2 compares the proposed architecture against the baselines regarding turn-taking and barge-in abilities. It can be observed that performance is comparable with and without the latent reasoning mechanism. Our method achieves better or comparable performance compared to non-thinking methods. Additionally, the MOS scores demonstrate the high quality of the audio synthesized by our method. Table 3 presents the results on Full-Duplex-Bench (Lin et al.,

*Table 3.* Turn-taking and barge-in performance on Full-Duplex-Bench. Results of baseline systems are taken from (Lin et al., 2025). We use "thk" to denote the proposed latent reasoning.

| Method | Turn-taking | | Barge-in | | |
|---|---|---|---|---|---|
| | TOR ($\uparrow$) | Latency ($\downarrow$) | TOR ($\uparrow$) | GPT-4o ($\uparrow$) | Latency ($\downarrow$) |
| Freeze-Omni (Wang et al., 2024b) | 33.6 | 0.95 | 86.7 | 3.62 | 1.41 |
| dGSLM (Nguyen et al., 2023) | 97.5 | 0.35 | 91.7 | 0.20 | 2.53 |
| Moshi (Défossez et al., 2024) | 94.1 | 0.27 | 100 | 0.77 | 0.26 |
| Gemini Live [1] | 65.5 | 1.30 | 89.1 | 3.38 | 1.18 |
| FLAIR w/o $thk$ | **94.1** | **0.37** | 89.0 | 4.08 | **0.35** |
| FLAIR w/ $thk$ | 93.0 | 0.43 | **92.0** | **4.22** | 0.36 |

[1] https://ai.google.dev/gemini-api/docs/live

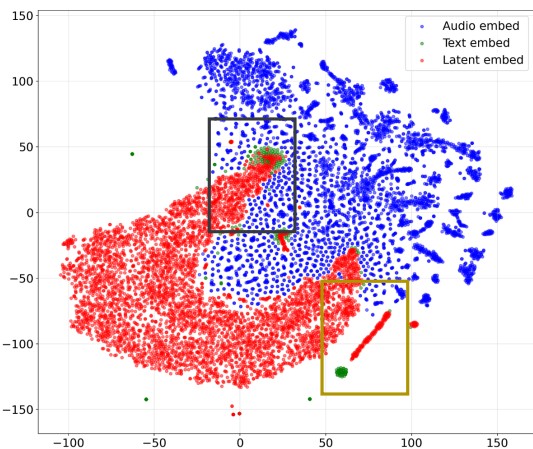

*Figure 3.* The distribution of the input audio, target text, and latent reasoning embeddings. Specifically, the latent reasoning embeddings act as a bridge that connects the input audio with the target text.

2025). Given that the CANDOR dataset used for turn-taking evaluation is a real-world noisy dataset, we construct an additional training subset by linearly mixing background noise ranging from 0 dB to 60 dB. Further details are provided in Appendix C. It can be observed that, following this data augmentation, our method remains robust in real-world noisy scenarios, achieving both high TOR and low latency. Performance on barge-in tasks also surpasses or is comparable to non-thinking methods. The performance in turn-taking and barge-in remains unaffected regardless of the inclusion of latent reasoning, demonstrating that our method maintains the conversational capabilities of the full-duplex system while delivering higher-quality responses.

### 7.3. Discussion and Visualization

We conduct the ablation study on data combination and architecture, including the streaming speech encoder, the expert model, and the streaming speech synthesizer. More discussions are attached in Appendix E.

To visually demonstrate the mechanism by which latent reasoning facilitates response generation, we randomly sample 300 instances from the Llama Questions dataset and extract the embeddings for each frame of the user's speech, the target text response, and the latent embeddings generated by the LLM during the user's speech frames. These representations are projected into a two-dimensional space using t-SNE (Maaten & Hinton, 2008). The distribution is illustrated in Figure 3.

We observe that the latent embeddings function as a bridge connecting the user's input audio embeddings to the assistant's target text embeddings. The distribution exhibits a distinct trend: originating from the input audio embeddings, it navigates through the feasible space to establish a trajectory toward the target text embeddings. This phenomenon is particularly pronounced within the yellow box: when target embeddings lie outside the cluster of input embeddings, the latent reasoning process forms a direct trajectory to reach the destination. Similarly, in the black box, the distribution of latent embeddings departs from the starting point and charts the shortest path through the feasible manifold (the void regions).

## 8. Conclusion

In this work, we introduce FLAIR, a novel framework that empowers full-duplex SDLMs with the capability to perform latent reasoning concurrently with speech perception. By proposing an ELBO-based training objective which leverages a Global-aware Expert model, we successfully enable effective SFT without the need for explicit reasoning datasets. Extensive evaluations demonstrate that FLAIR significantly enhances response quality across diverse benchmarks, including factual QA and open-ended conversation, especially for tasks requiring reasoning. Crucially, our approach maintains the efficiency required for real-time interaction, achieving superior turn-taking and barge-in performance without introducing additional inference latency or computational overhead. FLAIR represents a pivotal step toward developing more natural, human-like spoken dialogue agents that can truly listen, think, and speak simultaneously.

## Acknowledgements

This research is supported by the National Research Foundation, Singapore under its National Large Language Models Funding Initiative. Any opinions, findings and conclusions or recommendations expressed in this material are those of the authors and do not reflect the views of National Research Foundation, Singapore.

## Impact Statement

This paper presents work whose goal is to advance the field of Machine Learning. There are many potential societal consequences of our work, none of which we feel must be specifically highlighted here.

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

## A. Evidence Lower Bound

Latent Variable Models (LVMs) seek to capture the underlying structure of observed data $x$ by assuming the existence of unobserved random variables $z$, referred to as latent variables (Bishop, 1998). In this framework, the data generation process is defined by sampling $z$ from a prior distribution $P(z)$, followed by generating $x$ from the conditional distribution $P(x|z)$. The primary objective in LVMs is to maximize the marginal likelihood $\log P(x)$. However, for most complex generative models, direct computation of this likelihood is intractable as it requires integrating over the high-dimensional latent space:

$$\log P(x) = \log \int P(x, z)dz = \log \int P(x|z)P(z)dz \tag{14}$$

Since the true posterior $P(z|x)$ is generally unavailable, Variational Inference (VI) introduces a parameterized variational distribution $q_\phi(z|x)$ to approximate it (Zhang et al., 2018). By applying Jensen's Inequality, ELBO is introduced as:

$$
\begin{aligned}
\log P(x) &= \log \int q_\phi(z|x) \frac{P(x, z)}{q_\phi(z|x)} dz \\
&\geq \int q_\phi(z|x) \log \left( \frac{P(x|z)P(z)}{q_\phi(z|x)} \right) dz \\
&= \mathbb{E}_{q_\phi(z|x)}[\log P(x|z)] - \mathrm{KL}[q_\phi(z|x)||P(z)]
\end{aligned}
\tag{15}
$$

The ELBO encapsulates two distinct optimization forces (Kingma & Welling, 2013; Higgins et al., 2017):

- Expected Reconstruction Likelihood: The term $\mathbb{E}_{q_\phi(z|x)}[\log P(x|z)]$ encourages the variational distribution $q_\phi(z|x)$ to assign high probability to latent states $z$ that can accurately reconstruct the observed data $x$. This ensures that the latent variables capture the essential features necessary for subsequent generation tasks.

- KL Divergence: The term $\mathrm{KL}[q_\phi(z|x)||P(z)]$ acts as a regularizer that forces the approximate posterior to remain close to the prior $P(z)$. This prevents model collapse and structures the latent space, providing robustness when dealing with sequential or uncertain inputs (Kingma & Welling, 2013).

## B. ELBO in Full-duplex SDLMs

By replacing the standard $P(X)$ with the conditional $P_\theta(Y^{txt}|X)$, the objective of ELBO can be formulated as:

$$
\begin{aligned}
\log P_\theta(Y^{txt}|X) &= \log \int P_\theta(Y^{txt}, Z|X)dZ \\
&= \log \int q_\phi(Z|X, Y^{txt}) \frac{P_\theta(Y^{txt}, Z|X)}{q_\phi(Z|X, Y^{txt})} dZ \\
&\geq \int q_\phi(Z|X, Y^{txt}) \log \left( \frac{P_\theta(Y^{txt}|Z, X)P_\theta(Z|X)}{q_\phi(Z|X, Y^{txt})} \right) dZ \\
&= \mathbb{E}_{q_\phi(Z|X, Y^{txt})}[\log P_\theta(Y^{txt}|Z, X)] - \\
&\qquad\qquad \mathrm{KL}[q_\phi(Z|X, Y^{txt})||P_\theta(Z|X)].
\end{aligned}
\tag{16}
$$

This formulation shifts the model's focus from data reconstruction to response generation. The resulting conditional ELBO consists of two components that drive the latent reasoning capability of the SDLMs:

**Conditional Reconstruction**. The first term encourages the model to generate a high-quality response $Y^{txt}$ given the observed user input $X$ and the latent reasoning $Z$. It ensures that the latent representation, following a posterior distribution $q_\phi(Z|X, Y^{txt})$ offered by a "Global-aware Expert" model, captures the essential semantic information required for dialogue completion.

**Variational Regularization**. The second term functions as a variational regularizer within the ELBO framework (Higgins et al., 2017). It acts as a constraint that forces the causal model's latent prior $P_\theta(Z|X)$ to align with a Global-aware Expert

model's posterior distribution $q_\phi(Z|X, Y^{txt})$. Specifically, $q_\phi(Z|X, Y^{txt})$ represents an approximate posterior derived by a non-causal expert model that leverages the full conversation context (including the future response $Y^{txt}$) to form global insights. Conversely, $P_\theta(Z|X)$ represents the learned prior generated by the SDLM in a strictly causal, chronological manner. By minimizing this KL divergence, we guide the SDLM to internalize reasoning capabilities within its latent space, effectively approximating the global hindsight of the Expert model.

## C. Data Details

**Model Diversity and Prompt Speaker Pool**. To prevent the model from overfitting to the distributional bias of any single architecture, we employ a heterogeneous mix of foundation models for both text and speech generation. The text generation engine leverages an ensemble of high-performance LLMs, including GPT-OSS-120b [1], Qwen2.5-72B-Instruct [2], and Llama3.1-70b-Instruct [3]. Similarly, the acoustic synthesis is driven by diverse TTS systems, specifically Chatterbox [4], Magpie-TTS [5], and Mooncast [6]. For voice cloning, we construct a massive prompt bank by aggregating all speech segments with a duration of 5–10 seconds from the LibriTTS (Zen et al., 2019), YODAS (Li et al., 2023), and Hifi-TTS (Bakhturina et al., 2021) corpora. In total, over 100k distinct speech segments spanning more than 20k unique speakers are utilized to ensure high acoustic variance in the generated training data.

**Speech continuation data** (530K hours). Leveraging massive text pre-training corpora (Su et al., 2025), we construct synthetic pseudo-dialogues by alternately assigning sentences from continuous text passages to the user and the agent. This process transforms static text into semantically coherent multi-turn interactions. To mimic the variability of natural speech patterns, we implement a stochastic segmentation strategy for turn length: a turn concludes after a single sentence with a probability of 80%, while subsequent sentences are appended with decaying probability. Furthermore, to maintain conversational cadence, a mandatory role switch is enforced if a turn exceeds a 200-word threshold. Following textual formatting, the data is synthesized at the turn level. We sample two distinct speaker profiles to role-play the user and agent, ensuring acoustic separation. The resulting audio segments are then temporally aligned and concatenated to form a synchronized, two-stream conversational track. Through this process, a single flat-text data point is transformed into a multi-turn, multimodal full-duplex conversational example containing both user and agent speech. Futhermore. the role of user and agent can be swapped to improve the data efficiency.

**Instruction-Following QA Data** (70K hours). We leverage a diverse ensemble of LLMs to generate QA interactions between user and agent, encompassing both single-turn and complex multi-turn scenarios. Single-turn QA data (10K hours) aims to cover wide topic and multi-turns data enable the agent to handle long conversation up to 4 minutes. In these simulations, the agent is explicitly prompted to adopt the persona of a polite, helpful assistant dedicated to resolving user queries. To mitigate content redundancy and maximize topical diversity, we condition the dialogue generation on distinct textual contexts (e.g., Wikipedia data [7]) that serve as the conversation's thematic anchor. This strategy ensures that the dataset covers a broad semantic space rather than converging on generic conversational patterns. Then the generated textual turns are converted into speech waveforms sequentially via our multi-speaker TTS pipeline.

Simulation of user interruption. To enable robust handling of barge-in scenarios, we introduce synthetic interruptions into the multi-turn training data. For agent responses exceeding 4 seconds in duration, we trigger a user interruption with a probability of 10%. The interruption onset is randomized to occur within the 20% to 80% interval of the agent's utterance, simulating realistic timing variance. Upon the detection of an interruption, we model the agent's reaction latency by appending an 8-token delay (approximately 0.64 seconds) to the agent's output stream, after which the agent is forced to predict an End-of-Sequence (EOS token). This mechanism compels the model to cease generation and immediately transition back into the latent reasoning state to process the incoming user speech. During inference, we force the audio channel to silence token when text channel get a EOS token.

**ASR-QA Data** (20K hours). To incorporate real-world acoustic characteristics, we utilize authentic speech segments from multiple open-source ASR corpora (as combined in NeMo ASRSET (Noroozi et al., 2024b)) as contexts. Specifically, we

---

[1]https://huggingface.co/openai/gpt-oss-120b

[2]https://huggingface.co/Qwen/Qwen2.5-72B-Instruct

[3]https://huggingface.co/meta-llama/Llama-3.1-70B-Instruct

[4]https://github.com/resemble-ai/chatterbox

[5]https://docs.nvidia.com/nemo-framework/user-guide/latest/speech_ai/magpietts.html

[6]https://github.com/jzq2000/MoonCast

[7]https://huggingface.co/datasets/wikimedia/wikipedia

follow (Noroozi et al., 2024a) to create additional QA pairs based on the ASR transcription, where an LLM is prompted to generate a relevant question based on the transcribed content. Although the knowledge density of these samples is constrained by the short duration of the source clips, the human speech with natural background noise can enhance the model's robustness against real-world acoustic variability in a full-duplex setting.

**Acoustic Augmentation**. To further bolster the model's resilience to environmental acoustics, we implement a rigorous data augmentation regimen. Beyond applying SpecAugment to the user speech features, we curate a repository of 10,000 distinct background noise clips sourced from Freesound (Fonseca et al., 2017) and MUSAN (Snyder et al., 2015). During the training phase, these noise signals are dynamically superimposed onto the user speech stream with an injection probability of 50%. The mixing intensity is varied by sampling a Signal-to-Noise Ratio (SNR) uniformly from the range of 0 dB to 60 dB.

The pipeline for dataset construction is released in (Artificial Analysis, 2026).

## D. Training Details

**Implementation details.** The details of model implementation follow (Artificial Analysis, 2026). We use NeMo Toolkit to implement and train the models (Kuchaiev et al., 2019) and all the models are trained on 64 A800 (80G) GPUs. The LLM backbone is initialized from the Qwen2.5-7B-Instruct (Team, 2024). A Parakeet-based encoder (600M) [8] is employed for speech encoder, which features a causal convolutional context to support streaming input and a Transformer-based modality adapter with 1024 hidden units to align audio features with the LLM's embedding space (Koluguri et al., 2025; Rekesh et al., 2023). The audio codec and the pretrained streaming flow-matching model follows (Du et al., 2024). Both the speech encoder and audio codec are frozen during training. The speech encoder and LLM operate at a frame rate of 12.5 Hz, while the audio codec runs at 25 Hz. Consequently, for each input frame (or hidden state) processed by the LLM, we predict two speech tokens to maintain temporal alignment.

For the Latent Reasoning SFT, the weights for the three loss functions are set to $\alpha = 3$ and $\beta = 5$. This specific configuration is selected based on extensive empirical evaluations across numerous hyperparameter combinations. Furthermore, to emphasize critical turn-taking states, we applied additional scaling factors to the reconstruction loss $\mathcal{L}_{reco}$ in $\mathcal{L}_{elbo}$ at specific text tokens: a 20-fold increase for the <BOS> token (marking the start of the assistant's speech) and a 10-fold increase for the <EOS> token (marking the end of the assistant's speech).

We use the AdamW optimizer with $\beta = (0.9, 0.98)$ and $weight\_decay = 0$. An inverse Square Root Annealing learning rate schedule is applied. The learning rate starts from $5e^{-4}$ with a warm-up of 2500 steps during the interleaved pre-training stage, while during the SFT stages, we set the learning rate as $5e^{-5}$. Training is performed in BFloat16 mixed precision. We apply gradient clipping with a threshold of 1.0 to prevent divergence.

## E. Ablation and Discussion

In this section, we share the data and architectural decisions and empirical findings derived during the development of FLAIR.

**Speech continuation data**. We posit that the Speech Continuation pre-training (PT) phase is pivotal for establishing the model's fundamental intelligence and interaction capabilities. Its significance is twofold. First, from a knowledge transfer perspective, while the semantic content encountered during this phase overlaps significantly with the text corpora seen during the LLM's original pre-training, the input modality fundamentally shifts from discrete text tokens to continuous speech embeddings. By exposing the model to large-scale, text-derived speech data, we effectively bridge the modality gap, mitigating the distribution shift between textual knowledge and acoustic perception. Second, from an interaction perspective, this phase serves as a behavioral sandbox. It familiarizes the model with the mechanics of full-duplex communication, training it to coordinate simultaneous listening and speaking processes and to manage turn-taking dynamics across a massive diversity of speakers. Interestingly, although the peer-to-peer nature of the continuation data may appear superficially *misaligned* with the objective of constructing a helpful assistant, our empirical results indicate that the pre-trained model spontaneously exhibits rudimentary instruction-following capabilities. Detailed performance metrics supporting this observation are presented in Table 4.

We observe that the model can respond the user spoken query in a MCQ format, although this kind of data never emerges

---

[8]https://huggingface.co/nvidia/nemotron-speech-streaming-en-0.6b

*Table 4.* Accuracy (%) of Pretraining model on QA benchmarks. "LlamaQ", "WebQ", and "OBQA" stand for Llama Questions, WebQuestions, and OpenbookQA respectively.

| Method | LlamaQ | WebQ | OBQA | MMSU |
|---|---|---|---|---|
| PT Model | 63.6 | 40.4 | 61.3 | 49.5 |

into training set. This capacity may stem from LLM's original SFT while our speech continuation data align the acoustic query to text, thus retaining the knowledge learn from text domain. Meanwhile, on open-ended test set like ApacaEval, the model mostly responds user query in only one sentence due to the length distribution in PT data. Instruction-Following QA Data would increase the response while still keep concise compared with LLMs' response. We provide a case study for illustration as shown in Table 5.

*Table 5.* Case study for PT model, FLAIR, and LLM (Qwen2.5-7B-Ins)

| User query | Can you tell me a very easy way to clean a shower head? |
|---|---|
| Response by PT model | A easy and effective method you can try at home is using common household items. |
| Response by FLAIR | Sure!  A simple way to clean a shower head is to remove it and soak it in white vinegar for about an hour.  This helps dissolve any mineral deposits or buildup.  After soaking, rinse it thoroughly with water. |
| Response by LLM | Certainly!  Cleaning a shower head is a straightforward process that can help improve water flow and remove mineral buildup. Here's a simple method:\n\nMaterials Needed:\nWhite vinegar\nPlastic bag\nRubber bands or clips\nTowel\n\nSteps:\nRemove the Shower Head:\n\nTurn off the water supply to the shower.\nUnscrew the shower head from the pipe.  You may need a wrench or pliers for this step.\nSoak the Shower Head:\n\nPlace the shower head in a plastic bag.\nPour enough white vinegar into the bag to cover the shower head.\nSecure the bag around the shower head with rubber bands or clips to keep the vinegar in contact with the shower head.\n\nLet It Soak:\n\nLeave the shower head in the vinegar for at least 8 hours, or overnight for best results.  The longer it soaks, the more effectively it will remove mineral buildup.\n\nRinse and Reattach:\n\nAfter the soaking period, remove the shower head from the bag.\nRinse the shower head thoroughly with water to remove any remaining vinegar.\nReattach the shower head to the pipe and turn the water supply back on.\nTest the Flow:\n\nTurn on the shower and check the water flow.  You should notice an improvement in the water pressure and clarity.\n\nThis method is effective and easy to do, making it a great way to keep your shower head clean and functioning well. |

**Streaming Speech Encoder**. We experimented with pre-trained streaming encoders ranging from 120M [9] to 600M [10] parameters, where the context window keep as same as their ASR default configuration. Our results reveal a critical sensitivity to the dimensional alignment between the encoder and the LLM backbone. Specifically, when using a smaller 120M encoder, the significant discrepancy in hidden embedding space relative to the LLM degraded turn-taking performance, resulting in a response success rate of only 70.7% on the LLaMA-Q test set. Conversely, scaling up to a 600M encoder alleviated this bottleneck; a simple linear projection adapter was sufficient to achieve a 100% response success rate. This suggests that larger encoders provide more robust semantic representations that align more naturally with the LLM's latent

---

[9] https://huggingface.co/nvidia/stt_en_fastconformer_hybrid_large_pc
[10] https://huggingface.co/nvidia/nemotron-speech-streaming-en-0.6b

space.

**Global-aware Expert Model.** For the design of the non-causal Expert model, we compare a BERT-based architecture against T5 variants (Ni et al., 2022). Experiments utilizing the T5-Large [11] encoder yield performance metrics virtually identical to our BERT baseline. While upgrading to the T5-3B [12] encoder produce marginal gains, the computational overhead from the massive parameter increase is disproportionate to the benefits. Consequently, we retain the BERT-based design as the optimal trade-off between efficiency and effectiveness.

**Streaming Speech Synthesizer.** We evaluate two distinct synthesis paradigms: streaming Flow Matching and multi-layer Audio Codec generation as used in (Hu et al., 2025). As the text prediction is already stable when training speech predictor, both approaches demonstrated the capability to produce stable, high-fidelity agent speech. Subjective evaluations via Mean Opinion Score (MOS) indicate negligible perceptual differences between the two (4.2 v.s. 4.3). Furthermore, in a small-scale A/B preference test, the majority of evaluators rated the outputs as a tie, confirming that our latent reasoning framework is agnostic to the specific choice of the backend waveform generator.

---

[11]https://huggingface.co/google-t5/t5-large
[12]https://huggingface.co/google-t5/t5-3b

