# OpenReview forum: "The Silent Thought: Modeling Internal Cognition in Full-Duplex Spoken Dialogue Models via Latent Reasoning"
_ICML.cc/2026/Conference — ICML 2026 regular_

### Official Review · Reviewer_FwYb · 2026-03-03

**Soundness:** 3
**Presentation:** 3
**Significance:** 3
**Originality:** 3
**Overall Recommendation:** 5
**Confidence:** 2

**Summary:**

The authors introduce a new approach to full-duplex spoken dialog systems which they call FLAIR (Full-duplex Latent and Internal Reasoning). One of the goals of their approach is to allow the dialog agent to perform reasoning while the user is speaking to it. Unlike previous approaches, FLAIR does this by recursively updating a latent embedding while the user is speaking and autonomously deciding when to feed this vector to the LLM for a forward pass with tokens. They train the system using an ELBO-based objective that avoids the need for explicit reasoning annotations. Their experiments on a number of speech benchmarks find that this approach out performs many competing methods while maintaining the low latency required for such dialog systems.

**Compliance With Llm Reviewing Policy:**

Affirmed.

**Final Justification:**

The rebuttal addressed my main concerns, which were minimal and I maintained my original score. Additional details regarding my reasoning are in my original review.

**Key Questions For Authors:**

Will the code, models, and data for this work be released along with the paper?

**Limitations:**

yes

**Strengths And Weaknesses:**

Strengths

1. The paper is well presented and the approach is novel and intuitive.
2. The experiments seem fairly comprehensive across a number of speech benchmarks and find very positive results for the method.

Weaknesses

1. It would be interesting to see a comparison estimating the performance-per-flop or efficiency somehow but not necessary.
2. A number of hyperparameters regarding the method are not discussed. The authors mention extensive experiments that led to setting alpha \= 3 and beta \= 5 but these experiments are not discussed anywhere in the appendix.

General Comments

- Equation 4: I think the “q(z|x)” appearing expectation term should be a subscript? Not a big deal, just noting.
- Line 183 (left): “we do not directly adopt the previous hidden state as the input for the current step t.” – why not?
- During listening the model is continuously consuming audio and processing it. What is the rate of these thinking cycles?
- How robust is the method? Suppose there is an abnormally long pause between speech (say the user does not respond for several minutes). Will the method still work as intended? Is this scenario somewhere in the benchmarks conducted?

Additional Related Work (authors might find interesting)

- Levinson SC and Torreira F (2015) Timing in turn-taking and its implications for processing models of language. Front. Psychol. 6:731. doi: 10.3389/fpsyg.2015.00731

---

> ### Author Rebuttal · Authors · 2026-03-30
>
> We sincerely thank the reviewer for the highly positive evaluation and the constructive feedback.
>
> Weakness:
>
> Q1: It would be interesting to see a comparison estimating the performance-per-flop or efficiency somehow but not necessary.
>
> - A1: The "thinking" rate is strictly synchronous with the acoustic feature frame rate of the streaming encoder. Because it merely replaces the padding token of existing full-duplex LLMs with a latent embedding during the standard forward pass during inference, it introduces zero inference latency and negligible FLOP overhead compared to generating explicit discrete reasoning tokens.
>
> Q2: A number of hyperparameters regarding the method are not discussed. The authors mention extensive experiments that led to setting alpha = 3 and beta = 5 but these experiments are not discussed anywhere in the appendix.
>
> - A2: Regarding the hyperparameters, our experiments revealed no significant overall differences. While performance marginally improved on certain datasets, it slightly degraded on others, demonstrating that our method is not sensitive to hyperparameter choices.
>
> General Comments:
>
> Q1: Equation 4: I think the “q(z|x)” appearing expectation term should be a subscript? Not a big deal, just noting.
>
> - A1: Thank you for your suggestion. We will correct the typo in Equation 4 to a subscript.
>
> Q2: Line 183 (left): “we do not directly adopt the previous hidden state as the input for the current step t.” – why not?
>
> - A2: Utilizing the weighted sum of the vocabulary rather than the direct last hidden state stems from a fundamental architectural constraint of LLM. In LLMs, the input embedding space and the last layer hidden state space are typically decoupled, leading to a significant representational mismatch. Directly using hidden states as the next step's input embeddings exacerbates this issue [1,2]. Attempting to bridge this gap with extensive retraining often leads to overfitting or catastrophic forgetting [1].
>
>   Conversely, computing a weighted sum of all vocabulary embeddings acts as a bridge. This method effectively aligns the hidden state output space with the input embedding space. Such alignment structurally guarantees representation compatibility when the embedding is fed back as input. In [2], extensive empirical baseline evaluations are provided, supporting this theory.
>
> Q3: During listening the model is continuously consuming audio and processing it. What is the rate of these thinking cycles?
>
> - A3: The "thinking" rate is strictly synchronous with the acoustic feature frame rate (12.5Hz) of the streaming encoder, introducing no latency.
>
> Q4: How robust is the method? Suppose there is an abnormally long pause between speech (say the user does not respond for several minutes). Will the method still work as intended? Is this scenario somewhere in the benchmarks conducted?
>
> - A4: The full-duplex-bench utilized in our evaluation inherently comprises multi-turn conversations [3]. Within this benchmark, the user's audio stream track is pre-recorded and fixed, whereas the duration of the LLM's generated response is variable. Consequently, when the LLM ends its response, the user does not necessarily initiate the next turn immediately. This asynchronous alignment inevitably introduces pauses. Our highly competitive Takeover Rate (TOR) scores empirically demonstrate that when the user resumes speaking after these pauses, our model remains fully robust and smoothly takes the turn.
>
>   To address this specific query, we conduct a targeted test where the user remains silent for 2 minutes. The results confirm that when the user eventually resumes speaking, the agent successfully maintains its context and responds normally. However, it is worth noting that current standard benchmarks do not explicitly feature test cases with such multi-minute prolonged pauses.
>
> Additional Related Work (authors might find interesting):
>
> Thank you for your suggestion. This paper helps. I will include it in the reference.
>
> Key Questions:
>
> Q: Will the code, models, and data for this work be released along with the paper?
>
> - A: Yes, we will fully open-source our entire data generation pipeline, training code, and model weights.
>
> [1] Zhang, Z., He, X., Yan, W., Shen, A., Zhao, C., Wang, S., ... & Wang, X. E. (2025). Soft thinking: Unlocking the reasoning potential of llms in continuous concept space. arXiv preprint arXiv:2505.15778.
>
> [2] Zeng, B., Song, S., Huang, S., Wang, Y., Li, H., He, Z., ... & Lin, Z. (2025). Pretraining language models to ponder in continuous space. arXiv preprint arXiv:2505.20674.
>
> [3] Lin, G. T., Lian, J., Li, T., Wang, Q., Anumanchipalli, G., Liu, A. H., & Lee, H. Y. (2025). Full-duplex-bench: A benchmark to evaluate full-duplex spoken dialogue models on turn-taking capabilities. arXiv preprint arXiv:2503.04721.

---

> > ### Author Rebuttal · Reviewer_FwYb · 2026-04-02
> >
> > The authors have resolved my concerns. After reviewing their response to my comments as well as to the other reviewers, I maintain my original overall recommendation (5) to accept this paper.

---

> > > ### Author Response · Authors · 2026-04-03
> > >
> > > We are truly heartened and deeply grateful for your strong support and recognition of FLAIR. We are delighted that our clarifications have fully addressed your concerns. We remain fully open to any further discussion or suggestions.
> > >
> > > Given that our discussion has reached such a positive conclusion, would it be possible for you to consider increasing your Confidence if you feel our rebuttal has further solidified your expert assessment of our work?

---

### Official Review · Reviewer_UomX · 2026-03-12

**Soundness:** 3
**Presentation:** 4
**Significance:** 4
**Originality:** 3
**Overall Recommendation:** 4
**Confidence:** 4

**Summary:**

The paper introduces FLAIR, a full-duplex spoken dialogue model that performs latent reasoning concurrently with speech perception. Instead of predicting meaningless silence tokens during the user's speaking phase, FLAIR recursively feeds soft, continuous embeddings derived from the LLM's output logits back as input, enabling implicit "think-while-listening" without violating causality or introducing inference latency. To supervise this latent reasoning without explicit annotations, the authors design an ELBO-based training objective that employs a non-causal Global-aware Expert model to derive approximate posterior embeddings, which guide the causal LLM via reconstruction and KL-divergence losses. Experiments show that FLAIR improves response quality across a range of QA benchmarks — particularly on reasoning-heavy tasks — while preserving competitive turn-taking and barge-in performance comparable to existing full-duplex systems.

**Compliance With Llm Reviewing Policy:**

Affirmed.

**Final Justification:**

My concerns have been partially addressed. I will keep my score for supporting the acceptance of this paper.

**Key Questions For Authors:**

- How does the proposed method perform in multi-turn conversations? As a full-duplex speech LLM, multi-turn dialogue is likely one of its primary application scenarios.
- This further raises a question regarding the algorithmic implementation. In a multi-turn dialogue scenario, what context or information should the Global-aware Expert be exposed to?

**Limitations:**

No. The authors have not adequately discussed the limitations of their work.

**Strengths And Weaknesses:**

Strengths:

- This work introduces latent reasoning to replace the conventional paradigm in full-duplex speech LLMs where only a single token is produced during the listening stage. The proposed approach improves response quality while eliminating the need for additional annotations during the listening phase.

- This approach does not introduce additional latency and also removes the need to determine when the model should begin explicit reasoning.

Weaknesses:

- I remain somewhat skeptical about the claimed improvement in response quality brought by this method. In several prior works [1, 2] on latent reasoning, token-level reasoning is often used to guide the learning of latent reasoning representations. In contrast, the proposed method relies solely on the ELBO loss for training. Therefore, it is unclear whether the learned latent embeddings are genuinely related to improving response quality, or if the model is simply capturing certain patterns in the training data.

[1] Tan et al., "Think Silently, Think Fast: Dynamic Latent Compression of LLM Reasoning Chains", in NeurIPS, 2025.

[2] Gozeten et al., "Continuous Chain of Thought Enables Parallel Exploration and Reasoning", in ICLR, 2026.

---

> ### Author Rebuttal · Authors · 2026-03-29
>
> We sincerely thank the reviewer for the constructive feedback and the positive evaluation of our work. Below, we directly address the specific questions regarding the empirical validity of the ELBO-guided embeddings and the design for multi-turn scenarios.
>
> Weakness：
>
> Q：I remain somewhat skeptical about the claimed improvement in response quality brought by this method. In several prior works [1, 2] on latent reasoning, token-level reasoning is often used to guide the learning of latent reasoning representations. In contrast, the proposed method relies solely on the ELBO loss for training. Therefore, it is unclear whether the learned latent embeddings are genuinely related to improving response quality, or if the model is simply capturing certain patterns in the training data.
>
> A：While prior works distill representations from explicit CoT tokens, our method enforces reasoning through the Global-aware Expert. By accessing the future target response, the expert generates a globally-informed posterior. The KL-divergence loss forces the LLM to approximate this posterior. Thus, the latent embeddings are rigorously optimized to encode semantic planning for the response, rather than capturing superficial data patterns. The significant gains specifically on reasoning-heavy benchmarks (e.g., MMSU in Table 1) empirically validate this genuine reasoning capability.
>
> Key Questions：
>
> Q1: How does the proposed method perform in multi-turn conversations? As a full-duplex speech LLM, multi-turn dialogue is likely one of its primary application scenarios.
>
> A1: Existing multi-turn speech benchmarks rely on pre-recorded user audio. In a generative full-duplex setting, the agent's responses are open-ended and unpredictable. Pre-recorded user audio cannot dynamically adapt to the agent's specific replies, breaking the logical coherence of multi-turn interactions. In this paper, we evaluate our system on standard benchmarks universally adopted by current full-duplex works [1,2,3]. Nevertheless, our architecture inherently supports multi-turn conversations. We include a real-world multi-turn example in the revised paper to demonstrate this capability.
>
> Q2: This further raises a question regarding the algorithmic implementation. In a multi-turn dialogue scenario, what context or information should the Global-aware Expert be exposed to?
>
> A2: Our training samples are formatted as multi-turn dialogues. In our implementation, the Global-aware Expert processes the complete, unsegmented multi-turn dialogue sequence simultaneously. As formulated in Eq.5, the objective of the Expert model is to approximate the posterior distribution. Within a continuous, unsegmented multi-turn dialogue sequence, the optimal latent variable depends not only on the historical context (how it bridges the preceding dialogue) but also on the future (how it guides the trajectory of subsequent turns). If the sequence is artificially truncated at the current turn, the Expert degenerates into a locally-restricted model, rendering it incapable of deriving a mathematically true "Globally Optimal Posterior". Consequently, utilizing the full sequence as input is a theoretical necessity for computing this optimal posterior.
>
> [1] Yu, W., Wang, S., Yang, X., Chen, X., Tian, X., Zhang, J., ... & Zhang, C. (2025). Salmonn-omni: A standalone speech llm without codec injection for full-duplex conversation. arXiv preprint arXiv:2505.17060.
>
> [2] Hu, K., Hosseini-Asl, E., Chen, C., Casanova, E., Ghosh, S., Żelasko, P., ... & Ginsburg, B. (2025). SALM-Duplex: Efficient and Direct Duplex Modeling for Speech-to-Speech Language Model. arXiv preprint arXiv:2505.15670.
>
> [3] Défossez, A., Mazaré, L., Orsini, M., Royer, A., Pérez, P., Jégou, H., ... & Zeghidour, N. (2024). Moshi: a speech-text foundation model for real-time dialogue. arXiv preprint arXiv:2410.00037.

---

> > ### Author Rebuttal · Reviewer_UomX · 2026-04-04
> >
> > Thank the authors for their rebuttal. However, I still have concerns about the performance in multi-turn scenarios, and I believe clearer performance evidence is needed to support the authors’ claims.

---

> > > ### Author Response · Authors · 2026-04-04
> > >
> > > Thanks for your kind reply. To fully address your concern, we performed a **multi-turn** evaluation using Full-Duplex Bench 2.0 official code (https://github.com/DanielLin94144/Full-Duplex-Bench/tree/main/v2). It is noted that:
> > >
> > > - It employs GPT-realtime as a simulated examiner to conduct a multi-turn conversation.
> > > - The GPT examiner asks the full-duplex agent to start a daily chat or complete certain tasks.
> > > - Such an "online examiner" adjusts its requests based on the agent's responses, thereby addressing the context issues caused by pre-recorded multi-turn user speech.
> > > - The official code was released after the ICML deadline, so we did not include this evaluation in the paper.
> > >
> > > ---
> > >
> > > ### Results on **Daily** subset
> > >
> > > The examiner typically conducts a 2-minute conversation (4–8 turns, depending on the agent’s response length). The evaluation metrics are **TT** (turn-taking score, 1–5) and **IF** (instruction-following score, 1–5), where the scoring criteria follow the official code.
> > >
> > > | Model       | TT   | IF   |
> > > |------------|------|------|
> > > | FreezeOmni | 3.14 | 2.34 |
> > > | Moshi      | 3.73 | 2.67 |
> > > | Ours       | **3.80** | **3.07** |
> > >
> > > ---
> > >
> > > ### Results on **Entity Tracking** subset
> > >
> > > In addition to TT and IF, this subset includes a task-specific metric (**ET**, entity tracking) to evaluate the overall conversational quality.
> > >
> > > | Model       | TT   | IF   | ET   |
> > > |------------|------|------|------|
> > > | FreezeOmni | 3.49 | 2.44 | 2.62 |
> > > | Moshi      | **3.84** | 2.66 | 2.76 |
> > > | Ours       | 3.81 | **3.12** | **3.06** |
> > >
> > > ---
> > >
> > > We believe these results demonstrate that our model can handle **multi-turn** scenarios effectively: the turn-taking ability is comparable to Moshi, while other capabilities significantly surpass the baselines.

---

### Official Review · Reviewer_77F9 · 2026-03-12

**Soundness:** 2
**Presentation:** 2
**Significance:** 2
**Originality:** 3
**Overall Recommendation:** 2
**Confidence:** 3

**Summary:**

This paper introduces FLAIR, a novel framework for enabling "think-while-listening" capabilities in full-duplex Spoken Dialogue Language Models (SDLMs). The authors propose a latent reasoning mechanism where the model performs implicit, continuous thinking via dense embeddings, trained by an Evidence Lower Bound (ELBO) objective. A non-causal "Global-aware Expert" model is used during training to derive ideal latent reasoning paths from full dialogue context, which then supervises the causal streaming model via a KL divergence loss. Results show that FLAIR improves response quality, particularly on reasoning-heavy tasks, while maintaining the low latency and high barge-in success rates required for real-time interaction.

**Compliance With Llm Reviewing Policy:**

Affirmed.

**Final Justification:**

The author clarifies part of the concerns, while some other concerns remain, as mentioned in the last note. The score is adjusted, but still inclined to reject it.

**Key Questions For Authors:**

- Add a figure which visualizes the model architecture and input-output flows.
- Add the ablation study.
- More explanation and justification on the logic of the timing predictor and global-aware expert.
- Provide more empirical verifications to indicate the effectiveness of the thinking mode.

**Limitations:**

yes

**Strengths And Weaknesses:**

**Strength**
- Novel and Important Problem Definition: The authors strive to present a significant problem: how to utilize the "dead air" time in full-duplex dialogue. Framing this as a latent reasoning challenge, rather than a trivial silence-prediction task, is both insightful and timely for the advancement of conversational AI.
- Theoretically Grounded Methodology: The use of variational inference and the ELBO to train a causal model with a non-causal "teacher" is well-motivated. Its derivation provides a solid theoretical foundation for the proposed SFT strategy.
- Empirical Improvements: The performance gains, especially on reasoning tasks like MMSU and LlamaQ, are convincing.

**Weakness**
- The modeling training framework is not introduced clearly, and the current methodology seems to be problematic.
(1) Although Figure 2 provides some necessary information, it is more like a pipeline indication, while the model architecture visualization is lacking. Currently, it is solely introduced verbally in Section 4, which is hard to understand.
(2) The paper uses a "timing indicator" which predicts whether to respond or not. However, in dual-duplex systems, such a decision is inevitably affected by the user's (maybe multiple users') streaming speech, which basically is **not predictable**, from my point of view.
(3) The open-ended conversations have widespread topics and divergent distributions. It is also difficult to believe that a pretrained "global-aware expert" can be adequate to capture the posterior of latent, and can be generalizable enough in the inference stage. More explanations are needed.
- The paper's writing and organization need to improve:
(1) The first and third major contributions seem to be similar, and it might be better to merge them.
(2) The first and last paragraphs of Section 4.1 reclaim the benefits of ELBO, which have been discussed similarly in the preliminary knowledge of ELBO (Section 3.2). Maybe remove one of them.
- Some technical details are mismatched or unclear. For example,
(1) Section 3.1 first defines h^l_t, then Eq.2 introduces a "h", without a corresponding definition.
(2) Eq.3 also proposes a series of R_{i, t} while "i" is not defined.
(3) There should be a clear definition of "z" when Section 3.2 introduces it.
(4) Eq.6 and Eq.9 use a "+" sign without further introduction. Should it mean a concatenation instead of addition (i.e., use \oplus)?
(5) Last term in Eq.5 is P(Z|X), which should be the prior distribution of **latent**. However, this term in Eq.11 becomes Softmax(y), where y is the output text tokens, not latent.
(6) Should the 'speaking' phase on page 273 be the 'listening' phase?
(7) In Table 1 and Table 3, the method names are 'w/o thk' and 'w/ thk'; in Table 2, they are 'w/o latent thk' and 'w/ latent thk'. Do they have the same meaning?
- Theoretical derivations are trivial. While ELBO is useful and helpful, it is also well-known and well-established. Although the paper uses its conditional probability version, the difference is minor. Appendix B might be enough to provide the lower bound, while Appendix A might be omitted (and add a direct citation).
- Some important claims or assumptions are not justified by experiments:
(1) The first paragraph of Section 4 claims that the current method is better than utilizing the last hidden state directly. However, there is no empirical comparison to verify it.
(2) I can not find an ablation study, neither in Section 7.3 nor in Appendix E.
- Key issues of current experiment results:
(1) Although w/ think apparently outperforms w/o think in Table 1,  it is not significantly better than w/o think at all in Table 2 and Table 3. Since the core motivation is to conduct the reasoning during the listening phase, this indifferences make the paper's basic claim questionable.
(2) In Table 1, Kimi-Audio is actually better than FLAIR on some benchmarks like SDQA, AlpacaE, OBQA, and MMSU. The bold numbers are misleading.
(3) The authors claim that the latent embedding bridges the audio input and the text output semantically, as shown in Figure 3. However, the red points (latent embedding) are not labeled with the corresponding time steps, which makes this claim questionable, especially for the black box.
- Extreme Reliance on Synthetic Data: The model is trained on a staggering 620K hours of synthetic data generated by a complex pipeline involving multiple LLMs and TTS systems. While impressive in scale, this raises significant concerns about reproducibility, computational cost, and potential biases inherited from the teacher models. The paper does not sufficiently explore how the model would perform with less data or with real, organic conversational data.
- Lack of Human Evaluation: While the use of GPT-4o scores and automated metrics is standard, the paper would be significantly strengthened by a human evaluation of dialogue quality and naturalness. The MOS score for speech quality is provided, but there is no human assessment of whether the "thinking" actually makes the conversation feel more intelligent or natural to a user.

---

> ### Author Rebuttal · Authors · 2026-03-28
>
> We appreciate your suggestions. We would like to take this opportunity to clarify a few misunderstandings regarding our paper.
>
> Weakness：
>
> W1:(1) Fig.1
>
> The duplex model is a complex system in which each module relies on pretrained models. Individual submodule design is not our core contribution. Due to space limits, all implementation details are detailed in Appx D. We will fully open-source our code and add any further necessary clarifications to the revised appendix.
>
> (2) Timing indicator
>
> The decision of "whether to respond or not" constitutes the core functionality in all full-duplex systems, which has been fully verified. Our "timing indicator" serves as a **semantic VAD**, which is a mature and reliable technology. The mechanism is predictable. **Experimental results** also demonstrate that our method maintains excellent full-duplex conversational behaviors on both real-world and synthetic datasets.
> Consistent with the prevailing literature, this study focuses on scenarios without multi-user overlapping.
>
> (3) Global-aware expert.
>
> There appears to be a **misunderstanding** regarding the global-aware expert. The expert is utilized **only during the training phase**. By accessing global information, it generates optimal latent embeddings as high-quality supervision. The expert is jointly trained with the entire network to generate embeddings that are optimized to teach the LLM how to plan appropriate responses.
>
> Crucially, the global-aware expert is entirely **discarded** during inference. After training, the LLM has learned to generate appropriate latent embeddings on its own. Consequently, the model performs purely causally at test time.
>
> W2&4:
>
> Contributions 1 & 3: Contribution 1 targets the specific "thinking-while-listening" mechanism for **full-duplex** systems. Contribution 3 highlights our broader introduction of latent reasoning to Speech LLMs, an architecture generalizable to **half-duplex** settings.
>
> Sec 3.2/4.1 & Appx A/B: Sec 3.2 and Appx A cover standard ELBO preliminaries, whereas Sec 4.1 and Appx B specifically derive the conditional ELBO for full-duplex systems. We can revise this as per your suggestions
>
> W3:
>
> We have fixed (1)(3)(7) in the revised paper. For (2), the definition of i exists before Eq.3. For (4), the '+' is intended to denote element-wise addition.
> For (5), our formulations are consistent within the proposed framework. To clarify, y is defined as the **text logits instead of the text tokens** (Sec 4.2), and Z is derived as the expectation over vocabulary embeddings, weighted by Softmax(y). The variational regularization loss is formulated as the KL divergence between Softmax(y) and the distribution obtained in Eq.7.
>
> W5:(1) Last hidden states.
>
> The advantages of utilizing the weighted sum of the vocabulary over the last hidden state have been validated by prior literature, as referenced in Sec 4. The reason is due to the representation distribution mismatch. You can refer to our response to reviewer FwYb (General Comments: Q2), due to the character limitation for rebuttal. Re-verifying this conclusion falls outside the scope of this paper.
>
> (2) Ablation study
>
> We would like to clarify that the ablation studies are presented in Appx E. We provide the analysis of the pre-training stage and the key submodules. Should the reviewer deem additional ablation studies on any other specific components necessary, we would be happy to include them in the revised manuscript.
>
> W6:
>
> As per Sec 3.1 & 7.2, our goal is enhancing **response quality** (Table 1) without sacrificing real-time interaction (Tables 2 & 3). Latent reasoning is designed for content, not interaction mechanics. The "indifference" in Tables 2 & 3 proves our method achieves quality gains with zero latency overhead or interaction degradation. We fix the bold number in the revised paper.
>
> Fig.3 illustrates semantic distribution, not a temporal sequence. As human reasoning is intrinsically non-linear, semantic information does not strictly align with chronological steps. The demonstrated "bridge" is distributional instead of chronological. We will clarify this in the caption.
>
> W7:Data
>
> We will open-source our data generation pipeline. Collecting high-quality, real full-duplex audio is cost-prohibitive. **Exploring "less data" contradicts the fundamental scaling laws of the research area of LLMs**. Our evaluations on **real-world datasets** confirm that our model generalizes robustly to non-synthetic scenarios.
>
> W8:Human evaluation
>
> In current LLM research, GPT score is a widely adopted and reliable proxy for human preference.  Speech quality is not our primary focus. Nevertheless, the provided MOS scores strictly confirm that latent reasoning achieves content improvements without degrading fundamental speech quality.
>
> Overall：
>
> Due to character limits, we have focused on addressing the most critical misunderstandings. Full references and supplementary materials will be provided in the subsequent discussion phase.

---

> > ### Author Rebuttal · Reviewer_77F9 · 2026-04-02
> >
> > Part of the weaknesses are not fully resolved. For example:
> >
> > - The unclear illustration of the model architecture and training mechanism.
> > - Reliance on solely synthetic data.
> > - Performance indifference with Kimi-Audio.
> > - w/ think does not help significantly on the benchmarks of Table 2 and 3.

---

> > > ### Author Response · Authors · 2026-04-03
> > >
> > > Thank you for the follow-up. We further clarify these specific points:
> > >
> > > - The unclear illustration of the model architecture and training mechanism.
> > >
> > > The architecture of our proposed method can be conceptually represented by three streams:
> > >
> > > The Always-On User Audio Stream:  The system continuously ingests the user's streaming audio. The speech encoder processes this audio into continuous acoustic embeddings at a fixed frame rate of 12.5 Hz. The acoustic embeddings are continuously fed into an LLM backbone at every step.
> > >
> > > The Agent Text/Latent Stream: Operating synchronously at 12.5 Hz, the LLM continuously incorporates the user audio stream alongside its own autoregressive state. This autoregressive state dynamically switches based on the timing indicator: (1) During the Listening Phase (Latent Reasoning): The LLM does not generate discrete text tokens or silence tokens. Instead, it generates a latent embedding from the generated text logits. The LLM's input at the subsequent step is the element-wise addition of the user audio embedding and this latent embedding. (2) During the Speaking Phase: The LLM predicts explicit text tokens. The input for the subsequent step becomes the addition of the user audio embedding and the text token embedding.
> > >
> > > The Agent Audio Stream: The speech generation module runs concurrently to synthesize the agent's voice. When the LLM is in the latent reasoning state, a silence token is passed to this stream, outputting silence.  When the LLM switches to the speaking state, the generated text tokens and hidden states drive the generation of speech semantic tokens, which are instantly converted into audio waveforms via a streaming flow-matching model.
> > >
> > > We are more than happy to **include a detailed algorithm pseudo-code block in the main text or appendix** to further enhance the clarity and the architecture design and the SFT pipeline.
> > >
> > > - Reliance on solely synthetic data.
> > >
> > > First, this represents a **factual misunderstanding** of our experimental setup. As documented in **Section 5.1 and Appendix C**, to incorporate real-world acoustic characteristics, our training data inherently includes **20K hours of ASR-QA data sourced from authentic, real-world ASR datasets**. Therefore, the reviewer's assertion that our pipeline relies solely on synthetic data directly contradicts the text of the manuscript.
> > >
> > > Second, as we have explained in our initial rebuttal and the paper, the ultimate efficacy of our method is **validated on real-world human speech datasets**. FLAIR demonstrates robust and competitive performance on human-speech benchmarks such as CommonEval and the noisy, real-world CANDOR corpus. Concluding that our work is bounded by synthetic data while ignoring our strong empirical results on real-world test sets is unfounded.
> > >
> > > - w/ think does not help significantly on the benchmarks of Table 2 and 3.
> > >
> > > We must respectfully reiterate the fundamental objective of our work, which has been **stated across the Abstract, Section 3.1, Section 7.2, and our initial rebuttal**.
> > >
> > > The explicit goal of FLAIR is to optimize the assistant's **output content quality** during the listening phase, while strictly **maintaining conversational behavior without degradation**. There is no theoretical basis or empirical expectation that introducing latent reasoning would improve interaction mechanics. Therefore, the "performance indifference" observed between the "w/o thk" and "w/ thk" settings in Tables 2 and 3 is **not a failure to improve**. Rather, it is **the exact validation of our method**. It empirically proves that FLAIR successfully executes complex continuous reasoning during the listening phase with absolute zero latency overhead and zero degradation to real-time interaction capabilities. We evaluate content improvements strictly in Table 1, and we evaluate the preservation of zero-latency interaction strictly in Tables 2 and 3.
> > >
> > > - Performance indifference with Kimi-Audio.
> > >
> > > First, as explicitly denoted in the **"FD" (Full-Duplex) column of Table 1, Kimi-Audio is a half-duplex model**, whereas FLAIR is a **full-duplex** model. Half-duplex systems process the complete user audio only after the user has entirely finished speaking. They are free from the severe computational and latency constraints required for real-time concurrent perception and barge-in. Consequently, they can afford unlimited reasoning time and generate much longer, unconstrained responses.
> > >
> > > Second, as we detailed in Section 7.1 and Appendix E of the manuscript, open-ended benchmarks (like AlpacaEval and CommonEval) exhibit a strong bias toward longer responses, which half-duplex models natively produce. FLAIR, designed for realistic, low-latency conversational dynamics, generates more concise and immediate responses.
> > >
> > > Penalizing a zero-latency full-duplex architecture for not matching one specific unconstrained half-duplex model contradicts the fundamental premise of full-duplex research.

---

### Decision · Program_Chairs · 2026-04-30

**Decision:**

Accept (regular)

**Comment:**

This paper describes a novel full-duplex dialogue method, FLAIR, that enables models to perform continuous latent reasoning while listening.
Reviewers appreciate the novelty of the work, the clear presentation (except one of the reviewers requested clarifications regarding the architecture), the proposed technically grounded approach, improvements demonstrated on benchmarks with reasoning, and the low latency. During the rebuttals, authors responded with clarifications regarding the approach and evaluation strategy and provided additional results, and responded to further questions. Reviewer UomX still has concerns about the performance in multi-turn scenarios, and requested clearer performance impact. It would be good to see the paper after an update with the new results for a better assessment.